# Selection for increased tibia length in mice alters skull shape through parallel changes in developmental mechanisms

Colton M Unger[1,2], Jay Devine[3], Benedikt Hallgrímsson[2,3,4], Campbell Rolian[2,5]*

[1]Department of Biological Sciences, University of Calgary, Calgary, Canada; [2]McCaig Institute for Bone and Joint Health, Calgary, Canada; [3]Department of Cell Biology and Anatomy, University of Calgary, Calgary, Canada; [4]Alberta Children's Hospital Research Institute for Child and Maternal Health, University of Calgary, Calgary, Canada; [5]Department of Comparative Biology and Experimental Medicine, Faculty of Veterinary Medicine, University of Calgary, Calgary, Canada

**Abstract** Bones in the vertebrate cranial base and limb skeleton grow by endochondral ossification, under the control of growth plates. Mechanisms of endochondral ossification are conserved across growth plates, which increases covariation in size and shape among bones, and in turn may lead to correlated changes in skeletal traits not under direct selection. We used micro-CT and geometric morphometrics to characterize shape changes in the cranium of the Longshanks mouse, which was selectively bred for longer tibiae. We show that Longshanks skulls became longer, flatter, and narrower in a stepwise process. Moreover, we show that these morphological changes likely resulted from developmental changes in the growth plates of the Longshanks cranial base, mirroring changes observed in its tibia. Thus, indirect and non-adaptive morphological changes can occur due to developmental overlap among distant skeletal elements, with important implications for interpreting the evolutionary history of vertebrate skeletal form.

*For correspondence:
cprolian@ucalgary.ca

## Introduction

Organismal development is a major determinant of phenotypic variation, and therefore is fundamentally related to how organisms evolve (*Hendrikse et al., 2007*; *Hallgrímsson and Lieberman, 2008*). Organisms are comprised of interrelated anatomical elements whose morphology is patterned by shared genetic pathways (i.e. pleiotropic genes) and often by the same developmental processes (*Hallgrímsson and Hall, 2005*; *Murren, 2012*). Shared genetic and/or developmental processes lead to morphological integration, that is, the tendency of sets of traits to covary more strongly internally than with traits in other sets (*Cheverud, 1996*). In turn, integrated individual anatomical structures contribute to the modular organization of biological systems (*Wagner et al., 2007*; *Hallgrímsson et al., 2009*).

If two anatomical structures are integrated due to underlying genetic phenomena, such as pleiotropy or linkage disequilibrium, then those traits are more likely to respond to selection in a concerted manner (*Armbruster and Schwaegerle, 1996*; *Cheverud, 1996*). As a result of integration, correlated responses to selection can result in phenotypic changes in some traits that are merely a consequence of covariation with other traits under selection (*Gould and Lewontin, 1979*; *Wagner, 1984*; *Price and Langen, 1992*; *Parsons et al., 2015*). Understanding how developmental processes lead to correlated responses to selection is pivotal to distinguishing adaptive changes from those that are non-adaptive, or potentially even maladaptive, in analyses of phylogeny, ancestral relationships and evolutionary change within lineages (*Gould and Lewontin, 1979*; *Riska, 1986*; *von Cramon-Taubadel, 2019*).

The bones of the terrestrial vertebrate cranial floor (basicranium) and the postcranial skeleton represent an interesting case of integration because they are physically distant, yet both develop by the process of endochondral ossification (*De Beer, 1937*; *White and Wallis, 2001*; *Mackie et al., 2008*). Endochondral ossification proceeds through the formation, expansion, and mineralization of a cartilaginous template, known as an anlage, that is patterned in utero and undergoes post-natal longitudinal expansion (*Kronenberg, 2003*; *Mackie et al., 2008*; *Lefebvre and Bhattaram, 2010*; *Roselló-Díez and Joyner, 2015*). In the limbs, ossification initiates by the formation of primary and secondary ossification centers and continues into post-natal development via specialized growth plates situated at the ends of the long bones (*Kronenberg, 2003*; *Mackie et al., 2008*; *Lefebvre and Bhattaram, 2010*). The postcranial growth plate, once formed, is comprised of three histologically distinct zones containing cartilage-producing cells (chondrocytes) in different physiological states: resting, proliferative, and hypertrophic (*Roselló-Díez and Joyner, 2015*).

The basicranium is comprised of three bones: the basioccipital, the basisphenoid, and the presphenoid, which make up the floor of the caudal portion of the skull in mammals and are formed by growth in the spheno-occipital and intersphenoidal synchondroses (*Wei et al., 2016*). Synchondroses are structurally analogous to growth plates, however, synchondroses grow bidirectionally and have duplicated proliferative and hypertrophic zones (*Wei et al., 2016*). Basicranial growth is thought to be a key determinant of overall skull shape. The basicranium is the first cranial skeletal element to develop and is controlled intrinsically by endochondral ossification-like mechanisms, whereas the face and calvarium are influenced by, and grow in response to, hormonal regulation of surrounding tissue and brain growth, respectively (*Scott, 1958*; *Waters and Kaye, 2002*; *Bastir and Rosas, 2006*; *Richtsmeier et al., 2006*). Additionally, the basicranium supports the brain and contains critical foramina for the passage of vasculature and cranial nerves and is therefore central to proper craniofacial development (*Lieberman et al., 2008*).

Several mouse models with mutations in key genetic regulators of endochondral ossification manifest convergent phenotypic changes in the limb skeleton and the basicranium, with similar underlying perturbations to the structure of their respective growth plates, for example *Evc1/2* (*Pacheco et al., 2012*), *Pthrp*, (*Amizuka et al., 1994*), *Fgfr2/3*, (*Chen et al., 1999*; *Yin et al., 2008*), *Ctnnb1* (*Day et al., 2005*; *Nagayama et al., 2008*). These models provide support for the idea that even distant anatomical structures such as the skull and limbs are to some degree morphologically integrated, through the action of pleiotropic genes. Whether this shared developmental architecture can also lead to correlated evolutionary change in these structures over time, however, is not known. Here, we used the Longshanks mouse to study correlated short-term evolution of cranial and postcranial skeletal elements. The Longshanks mouse was established through artificial selection for increased tibia length relative to body mass, using an outbred CD1 stock. By generation 20, mean tibia length in two independently selected Longshanks lines had increased by 13–15% in comparison to random-bred Controls from the same genetic background with no change in average body mass (*Marchini et al., 2014*; *Castro et al., 2019*).

Investigation of the cellular mechanisms governing limb development in Longshanks revealed structural alterations in the postnatal epiphyseal growth plate of the tibia. Specifically, the Longshanks selection regime resulted in larger tibial growth plates with larger resting and proliferative zones, without changes in cell division rate or timing of growth plate fusion compared to Controls (*Marchini and Rolian, 2018*). Previous analyses also suggested the tibia selection regime resulted in mice that are skeletally larger in relation to body mass, with correlated skeletal responses at the systemic level (*Sparrow et al., 2017*), along with potentially maladaptive changes in skeletal microarchitecture (*Farooq et al., 2017*; *Cosman et al., 2019*).

The Longshanks experiment offers a unique opportunity to study correlated evolution in skeletal traits that were not directly under selection, in a model with known evolutionary history, under controlled laboratory settings. Given the underlying developmental relationship between the long bones and cranial base, we investigated whether selection for increased tibia length indirectly altered the shape of the Longshanks cranium. We tested the general hypothesis that selection for increased tibia length produced indirect responses in the cranium of Longshanks through changes to the shared process of endochondral ossification. Specifically, we predicted that Longshanks crania will have a series of craniofacial morphological changes corresponding to altered synchondrosis size/architecture. To test this hypothesis, we compared the 3D shape of adult Longshanks crania from both Longshanks lines to Controls across three evenly spaced generations in the selection

experiment. We also used a combination of morphometric analysis and histology to investigate cranial development in Longshanks neonates.

## Results

### Longshanks adults

#### Body mass and cranium size allometry is altered in Longshanks adults

F01 mice (founders) that had not been subjected to selection did not differ in average weight or tibia length between lines (*Figure 1—figure supplement 1*, *Figure 1—source data 1*, *Supplementary file 1*). Moreover, random-bred F09 Controls and F20 Controls did not differ from F01 founders in terms of tibia length in ANCOVA accounting for body mass (*Figure 1—figure supplement 1*, *Supplementary file 1*). In contrast, LS1 and LS2 at F09 have an average of 7.3% longer tibiae compared to F09 Controls, while LS1 and LS2 at F20 have 16.4% longer tibiae on average when compared to F20 Controls (*Figure 1—figure supplement 1*, *Supplementary file 1*). Average body mass in our sample was stable between lines across all three generations and did not differ significantly in all but two pairwise comparisons between groups (ANOVA, F = 3.312) (*Figure 1—figure supplement 1*, *Supplementary file 1*). In contrast, at generation F20, LS1 and LS2 mice had significantly larger cranium centroid sizes than Controls (ANOVA, F = 312.7, Tukey's HSD, F20 LS1vsF20 CTL p<0.05, F20 LS2vsF20 CTL p<0.05), although the latter did not differ from F01 or F09 Controls (Tukey's HSD, F01 CTLvsF09 CTL p=0.929, F09 CTLvsF20 CTL p=0.803, F01 CTLvsF20 CTL p=1.000).

Given that the long bones of the Longshanks post-cranial skeleton are larger than Controls at any given body mass (*Sparrow et al., 2017*), we asked if the allometric scaling relationship between Longshanks crania and overall body mass had changed in response to 20 generations of selection. Body mass was significantly correlated with cranium centroid size in our adult sample (Pearson, r = 0.697, p<0.001). ANCOVA comparing mean cranium centroid size among lines using body mass as the covariate indicates that there was a significant difference in centroid size between Controls and Longshanks at F20; however, LS1 and LS2 did not differ from each other (F = 14.97, Tukey's HSD, LS1vsLS2 p=0.947, LS1vsCTL p<0.001, LS2vsCTL p<0.001). Hence, Longshanks selected lines have skeletally larger crania after 20 generations of selection independent of body mass (*Figure 1—figure supplement 2*).

#### The Longshanks cranium is longer, narrower, and flatter

Next, we asked if the fact that tibia length and cranium centroid size increases in F20 LS1 and LS2 is associated with shape differences in their cranium compared to F09 and F01 mice. We performed a principal component analysis on the residuals of a multivariate regression of shape on sex in order to control for potential sex effects on cranial shape in our sample. Comparison of principal component (PC) score means between groups demonstrates that despite overlap in skull shape, LS1 and LS2 have shifted substantially into positive PC1 space, reflecting crania that are longer and narrower with reduction in vault height (*Figure 1A*, *Figure 1—source data 1*, *Videos 1* and *2*). Post-hoc pairwise comparisons from a Procrustes ANCOVA comparing adult cranium shape by group, independent of sex effects, showed that all groups within line by generation, or within generation by line, differ in mean shape, except F01 LS2 and Controls (F = 10.679, F01 LS2vsF01 CTL p=0.203). When comparing the Euclidean distance among group sex-adjusted PC score means, however, F20 LS1 and LS2 mice are on average over twice as far from unselected mice in morphospace (i.e. all F01 founders, F09 and F20 controls) than the latter are from each other (mean Euclidean distances 0.023 vs 0.010, *Supplementary file 2*).

#### Longshanks cranial shape differences remain after controlling for skull size and sex

Given that PC1 generally captures differences in shape primarily due to allometric effects of size (*Klingenberg, 2016*), and that LS1 and LS2 have larger skulls in F20, we asked if the cranium of F20 selected lines score more positively simply because they are larger, and if larger skulls are associated with different cranium shapes in F20 selected lines compared to Controls and F01 unselected lines.

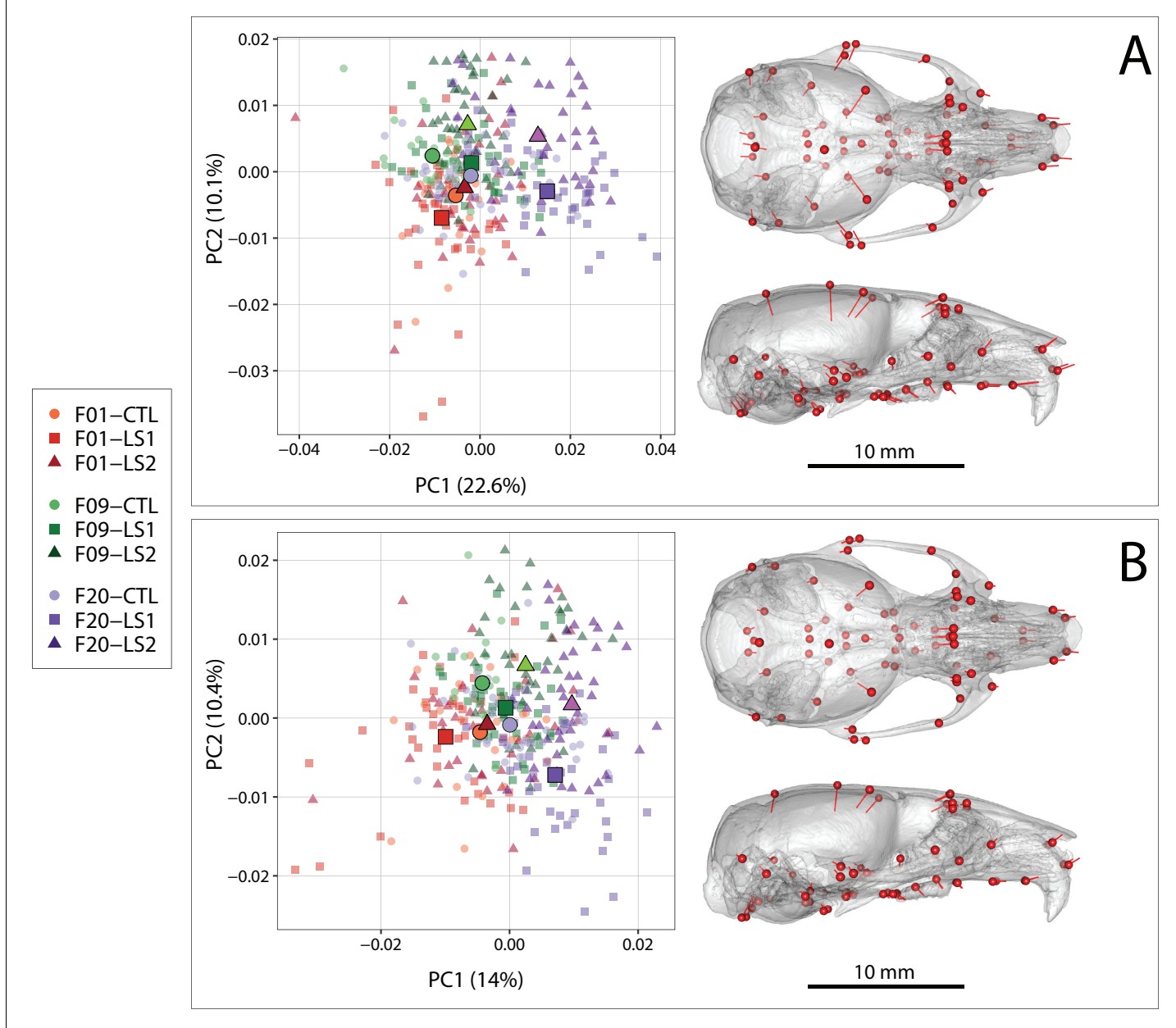

**Figure 1.** Scatter plots of the first two principal components (PC) of Procrustes shape variables in adult Longshanks and Controls throughout the selection process. (**A**) Plot of sex-adjusted Procrustes shape variables (left), and vectors of shape change at each cranium landmark (magnified two times for visualization) showing shape transformations along PC1 from negative to positive scores (right). Large symbols indicate mean PC1 and PC2 scores for each respective cohort. (**B**) Plot of Procrustes shape variables additionally corrected for size.

The online version of this article includes the following source data and figure supplement(s) for figure 1:

**Source data 1.** Adult morphometric and landmark data.
**Figure supplement 1.** Boxplots of adult Longshanks and Control metrics.
**Figure supplement 2.** Scatter plots with regression lines by group showing the relationships between body mass and cranium size (centroid size) in adult founder mice (F01) and after 20 generations of selection (F20).
**Figure supplement 3.** Scatter plot of fitted PC1 scores (shape scores predicted by regression of shape on size) vs log (centroid size) showing within group patterns of cranium allometry.
**Figure supplement 4.** Adult cranium landmarks used in this study in lateral, dorsal, and ventral landmark views.

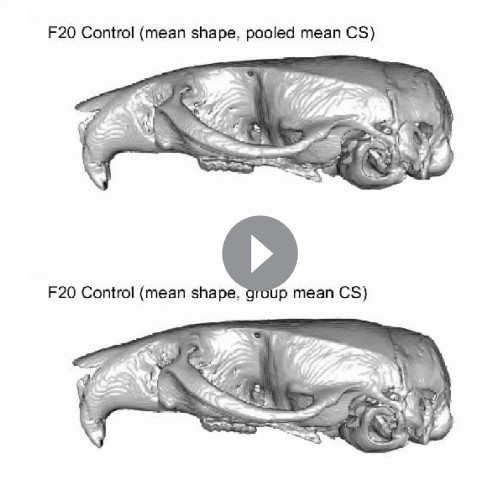

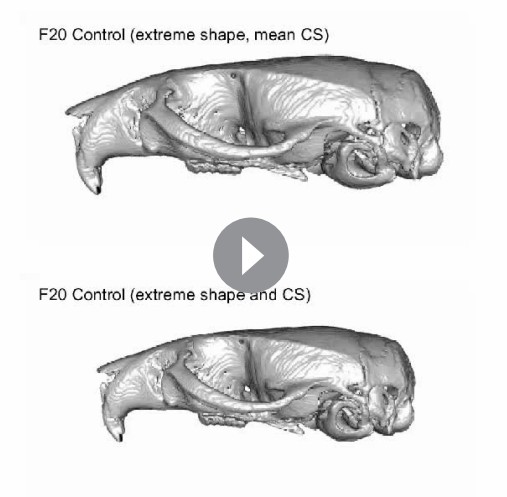

**Video 1.** Mean cranial deformation in Longshanks. Top: deformation of an F20 Control mean cranial shape into a F20 Longshanks Line 1 (LS1) mean cranial shape, with both mean configurations scaled to the pooled mean centroid size of the total sample (n = 327). Bottom: as above, but the skulls in each group have been scaled to the mean centroid size of that group. In both cases, the most significant changes relate to the flattening of the frontal and parietal regions, decreasing height of the braincase, and relative elongation of the snout in Longshanks.
https://elifesciences.org/articles/67612#video1

**Video 2.** Extreme cranial deformation in Longshanks. Top: deformation trajectory between one extreme Control male and one extreme LS1 male, selected from non-overlapping regions of morphospace (see *Figure 1*). The skull configurations are scaled to the pooled mean centroid size of the total sample (n = 327). Bottom: as above, but the two skulls have been scaled to those individuals' centroid sizes, revealing substantial differences in centroid size between these two individuals as well.
https://elifesciences.org/articles/67612#video2

We compared the fitted PC1 scores of a pooled within-group regression of shape on size to log (centroid size), which shows the cranium size and shape scaling relationship, between all nine groups in our sample (*Figure 1—figure supplement 3*). At any given centroid size, Longshanks F20 selected lines score more positively (longer and narrower) in predicted shape. Importantly, the slopes of the lines, which capture the scaling relationship between cranium shape and size did not differ significantly between any of the groups in our sample (p>0.05). Thus, while Longshanks F20 LS1 and LS2 have larger crania at any given body mass compared to Controls, the allometric pattern within the cranium itself was not altered by selection for increased tibia length.

The difference in intercept between the fitted PC1 scores and log centroid size of LS1 and LS2 in relation to Controls (*Figure 1—figure supplement 3*) suggest that while the increase in size of F20 selected crania contributes to the shape differences along PC1, it is not the only cause of shape variation. We therefore asked if differences in shape between the Longshanks and Control cohorts persist when the effect of size is removed from our sample by using multivariate regression residuals of shape on size. The principal component analysis of shape independent of size and sex shows a marked reduction in group separation along PC1; however, F20 LS1 and LS2 still typically score more positively in PC1, corresponding to crania that are relatively longer, narrower, and have reduction in vault height (*Figure 1B*). Post-hoc pairwise comparisons from a Procrustes ANCOVA comparing adult cranium shape by group independent of size and sex effects showed that all groups within line by generation, or within generation by line, differ in mean shape (F = 8.205, p<0.001) except F01 LS2 and Controls (F01 LS2vsF01 CTL p=0.173). As with sex-adjusted data, F20 Longshanks mice are substantially farther in morphospace from unselected groups than the latter are from each other (mean Euclidean distances 0.017 vs 0.010, *Supplementary file 2*).

## Tibia length and cranial shape means evolve under selection, but are uncorrelated within generations

After identifying underlying differences in cranial shape between F20 Longshanks and Controls irrespective of sex and size effects, we asked how this residual cranial shape variation was related to variation in tibia length, the principal selection target. To determine whether the residual cranial shape changes was a correlated response to tibia selection we performed within-group regressions on size and sex adjusted PC1 scores against tibia length (*Figure 2*, *Figure 1—source data 1*). Differences in the slopes of within-group regressions over time, and in comparison to Controls, would reveal if LS1 and LS2 crania experienced directional selection (*Price and Langen, 1992*). Clusters of points for Longshanks specimens progressively increase in tibia length and adjusted cranial shape score over time (*Figure 2A,B*), a trend which is not seen in Controls (*Figure 2C*). More importantly, within cohorts, tibia length and cranium shape are not significantly correlated (*Figure 2*). While the population means for residual cranium shape and tibia length were changing over time in Longshanks selected lines, selected individuals with relatively longer tibiae did not necessarily also have derived cranial shapes. In other words, unlike tibia length, residual cranial shape likely did not confer an advantage in terms of reproductive success.

## Intergenerational changes occurred in a stepwise process

F09 LS1 and LS2 score more positively along PC1 in the same direction as F20 LS1 and LS2 after removing size effects (*Figures 1B* and *3*, *Figure 1—source data 1*). In other words, F09 LS1 and LS2 appear to have intermediate shapes along PC1 between F01 groups and F20 selected lines (*Figure 3*). This led us to ask how intergenerational changes in cranium shape occurred throughout the selection process. We computed the mean shapes of LS1 and LS2 lines over time and compared them using deformation heat maps to track shape change between generations within a selection line. Our results show that indirect responses to selection in the Longshanks cranium occurred in a stepwise process: shape change in the first nine generations of selection contributed to the reduction in vault height, whereas the remaining 11 generations of tibia selection led to a reduction in cranial width at the zygomatic arches in parallel with snout elongation (arrows in *Figure 3*). In comparison, intergenerational changes in the Control lines shows virtually no change in cranial vault height between F01 and F09, and a reduction in the occipital area of the cranium from F09 to F20 (*Figure 3*).

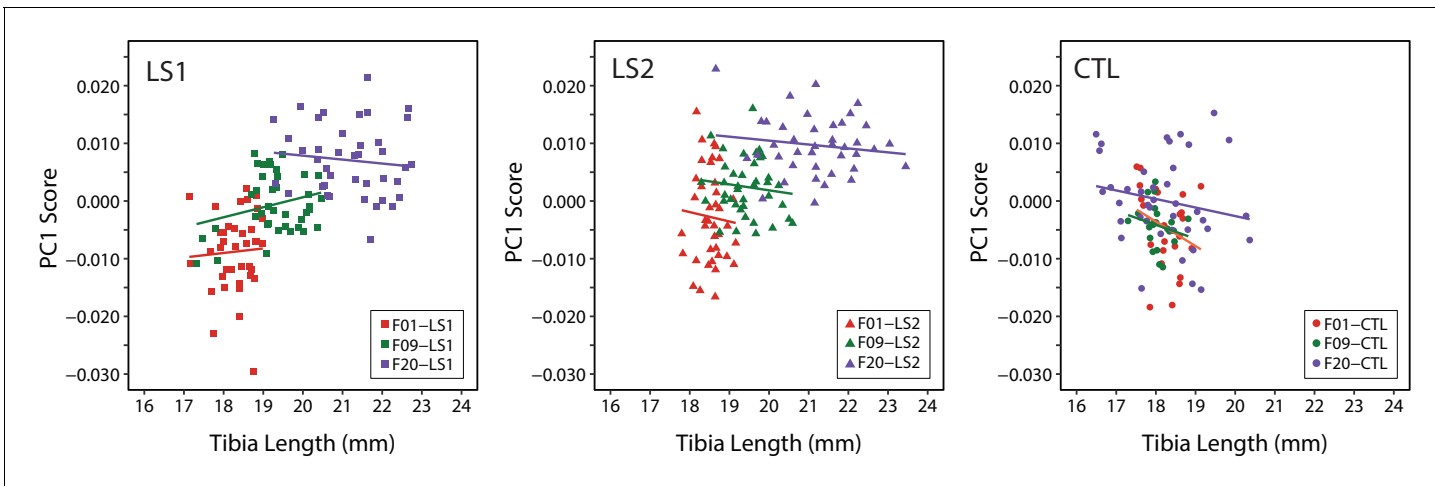

**Figure 2.** Scatter plots with regression lines by cohort showing the relationships between PC1 score (cranium shape) and tibia length in adult mice throughout selection. Cranium shape data is adjusted for the effects of sex and size related allometry.

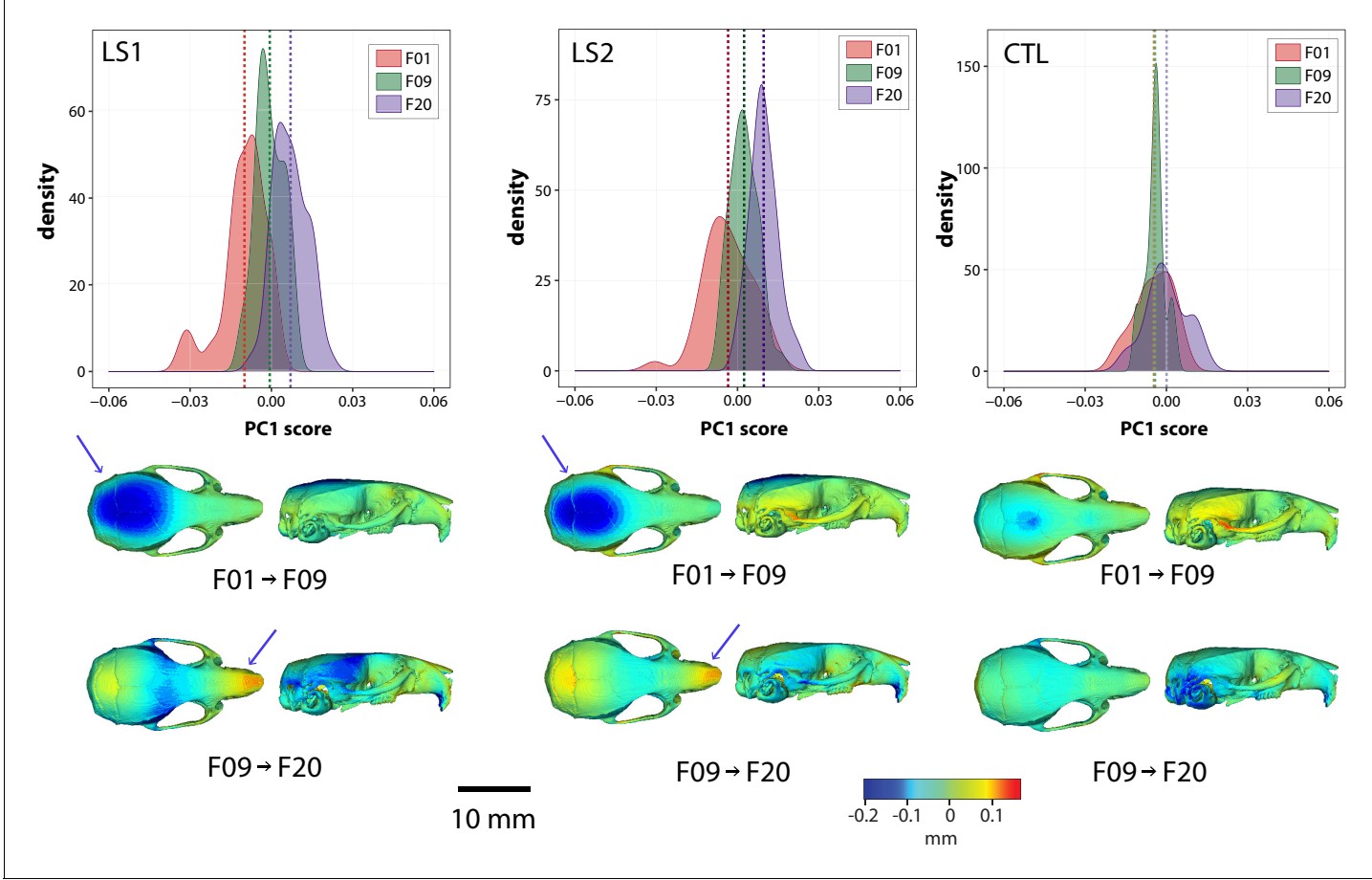

**Figure 3.** Intergenerational shape changes within both Longshanks lines and Controls throughout the selection process. Intergenerational shape changes within both Longshanks lines and Controls throughout the selection process. Top: Density plots following intergenerational shifts in mean PC1 scores within LS1 (left), LS2 (center), and CTL (right) lines for size and sex adjusted shape data. Bottom: Heatmaps showing shape transformations between mean shapes in the first 9 generations of selection (F01 to F09) and the next 11 generations (F09 to F20) after correcting for size and sex effects. Blue indicates areas of relative reduction, red indicates areas of relative expansion, and green indicates neutral areas. Longshanks independently undergo vault height reduction between generations F01 and F09, followed by snout elongation between generations F09 and F20 (blue arrows).

## Longshanks P07 neonates

### Neonate crania have similar shape patterns as Longshanks adults

We investigated if we could detect the adult pattern in shape differences earlier in ontogeny. Using one-week old (P07) Longshanks neonates, we compared cranial shape at a time when the Longshanks tibia is growing most rapidly and tibia length differences are already observable (*Farooq et al., 2017*; *Marchini and Rolian, 2018*; *Figure 4*, *Figure 4—source data 1*, *Supplementary file 3*). After regressing out litter size effects, our principal component analysis of neonate cranial shape showed a large separation in morphospace between the selected lines and Controls (*Figure 4*). LS1 and LS2 cluster more closely than Controls and have skulls that are longer, narrower and have reduced vault heights (*Figures 4* and *5*). The Procrustes ANCOVA and pairwise comparisons showed that LS1, LS2, and Control neonates significantly differ from each other in cranium shape (F = 14.395, LS1vsCTL p<0.001, LS2vsLS1 p<0.001, LS2vsCTL p<0.001). In addition, we observed via deformation heatmaps that the cranial pattern seen in Longshanks adults exists by one-week post partum and becomes more marked with age (*Figure 5*). The LS2 selection replicate appears to have reduced magnitudes of cranial response compared to LS1 at F20 and in neonates (*Figures 1B*, *3* and *5*).

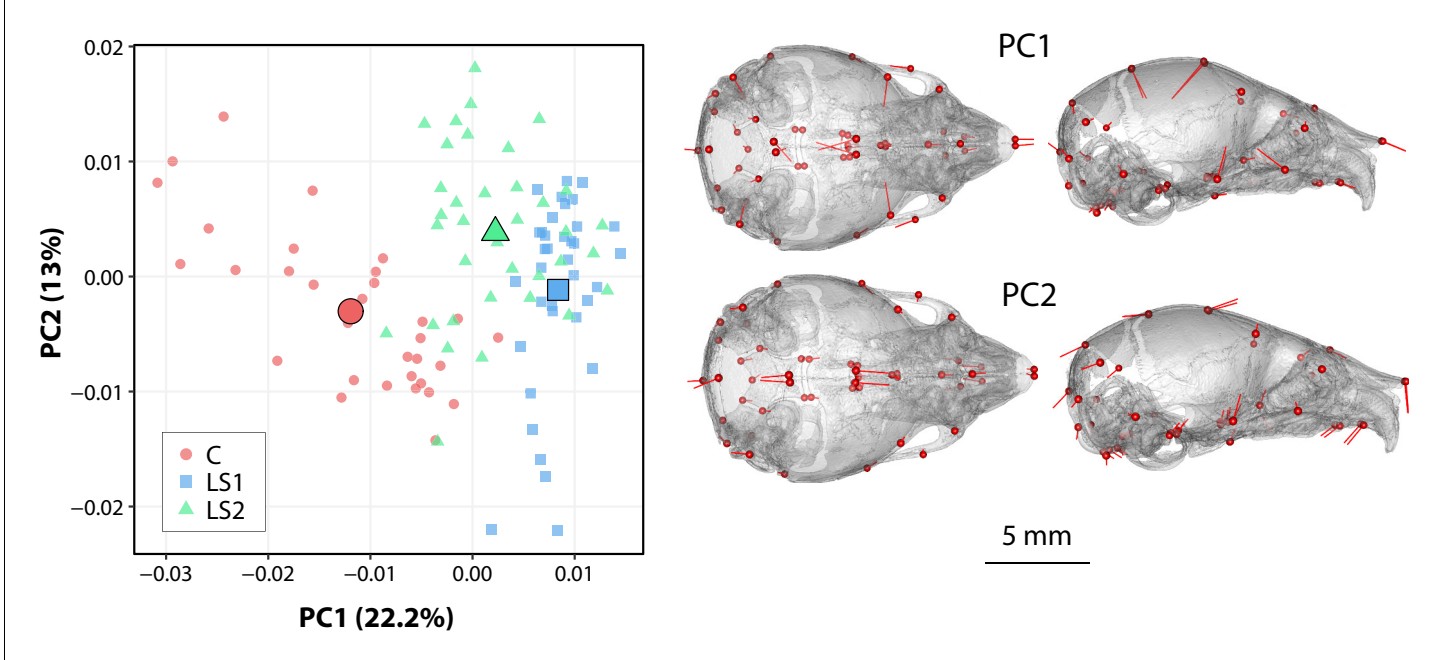

**Figure 4.** Scatter plots of the first principal components (PC) in neonate Longshanks and Control cranium Procrustes shape variables at generation 32 (F32). Left: Plot of litter size adjusted Procrustes shape variables (left), large symbols indicate mean PC1 and PC2 scores for each respective cohort. Shapes of individual points indicate Longshanks lines (circle = CTL, square = LS1, and triangle = LS2). Right: Neonate cranium with vectors of shape change at each cranium landmark (magnified four times for visualization) showing shape transformations along PC1 (top) and along PC2 (bottom) from negative to positive scores.

The online version of this article includes the following source data and figure supplement(s) for figure 4:

**Source data 1.** Neonate morphometric and landmark data.

**Figure supplement 1.** Boxplots showing differences in neonate Longshanks and Control metrics.

**Figure supplement 2.** Neonate (P07) cranium landmarks used in this study in lateral, dorsal, dorsal cranial base, caudal, and ventral landmark views.

### Longshanks neonate cranial bases are flatter than Controls and differ in synchondrosis shape

Given the underlying developmental relationship between the cranial base and the long bones, we asked if the neonate cranial bases differed in shape along the sagittal plane between Longshanks and Controls, where the synchondroses' primary axis of elongation exists. We performed a 2D morphometric analysis and found that groups differed in cranial base shape after removing litter size effects by Procrustes ANCOVA (F = 20.972, p<0.001). As with the neonate cranial form, Procrustes ANCOVA and pairwise comparisons of cranial base shape showed that LS1, LS2, and Control neonate mean cranial shapes all differ from each other. Longshanks neonate cranial base shapes differed from Controls in a similar pattern, but to different extents, with LS2 assuming an intermediate position in cranial base morphospace (*Figure 6A*, *Figure 6—source data 1*). Deformations comparing a mean Control cranial base to LS1 and LS2 means show a flattening of the cranial base in both Longshanks lines (*Figure 6B*). Moreover, the shape of the intersphenoidal synchondrosis changes in LS1 and LS2 compared to Controls, expanding dorsally to become more wedge-shaped, whereas the spheno-occipital synchondrosis shows no significant shape change differentiating them from the Control spheno-occipital synchondrosis.

shape (*Figure 6B*). This suggests that a cellular change in the dorsal aspect of the intersphenoidal synchondrosis could be driving cranial base flattening in the Longshanks juvenile skull, and hence potentially into adulthood given the broadly similar shape changes observed at both stages. To qualitatively validate the cellular changes inferred from our 2D geometric morphometric analysis, we collected and imaged cranial base sections near the sagittal midline. Since these developmental shape differences are still subtle, we selected two representative extreme specimens that had large

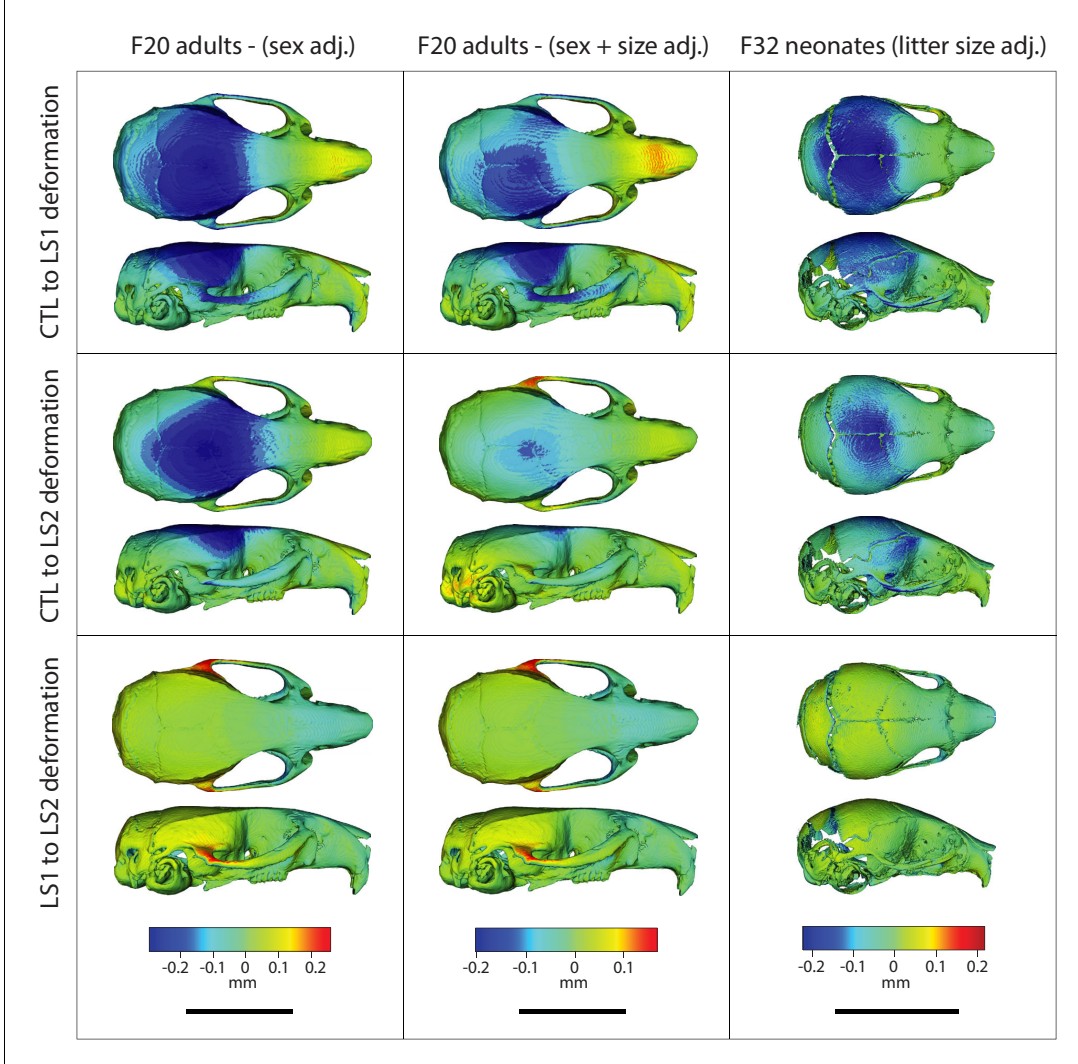

**Figure 5.** Comparison of adult and neonate cranial phenotypes through shape change heatmaps. Heatmaps show the deformations required to transform between the mean shape of a given cohort to the mean shape of another. Blue indicates areas of relative reduction, red indicates areas of relative expansion, and green indicates neutral areas. Scale bar = 10 mm.

differences in ossified tibia length, cranial base shape and cranium shape, yet comparable cranium centroid sizes so that size would not confound our analysis. In agreement with our morphometric data, the spheno-occipital synchondrosis does not differ qualitatively between these extreme specimens (*Figure 6C*). However, the intersphenoidal synchondrosis is markedly larger in our Longshanks specimen compared to the Control, with larger resting and proliferative zones that recapitulate the cellular differences characterized in the Longshanks epiphysis (*Figure 6C*; *Marchini and Rolian, 2018*). Crucially, the intersphenoidal synchondrosis is more wedge-shaped in our Longshanks specimen at the cellular level, supporting the observed intersphenoidal synchondrosis changes at the morphometric level (*Figure 6B,C*).

## Discussion

### Phenotypic changes

We investigated parallel skeletal size and shape changes in the limb and cranium in the selectively bred Longshanks mouse. Our morphometric analysis of adult cranium shape demonstrated that 20 generations of selection for longer tibiae relative to body mass are associated with the elongation of

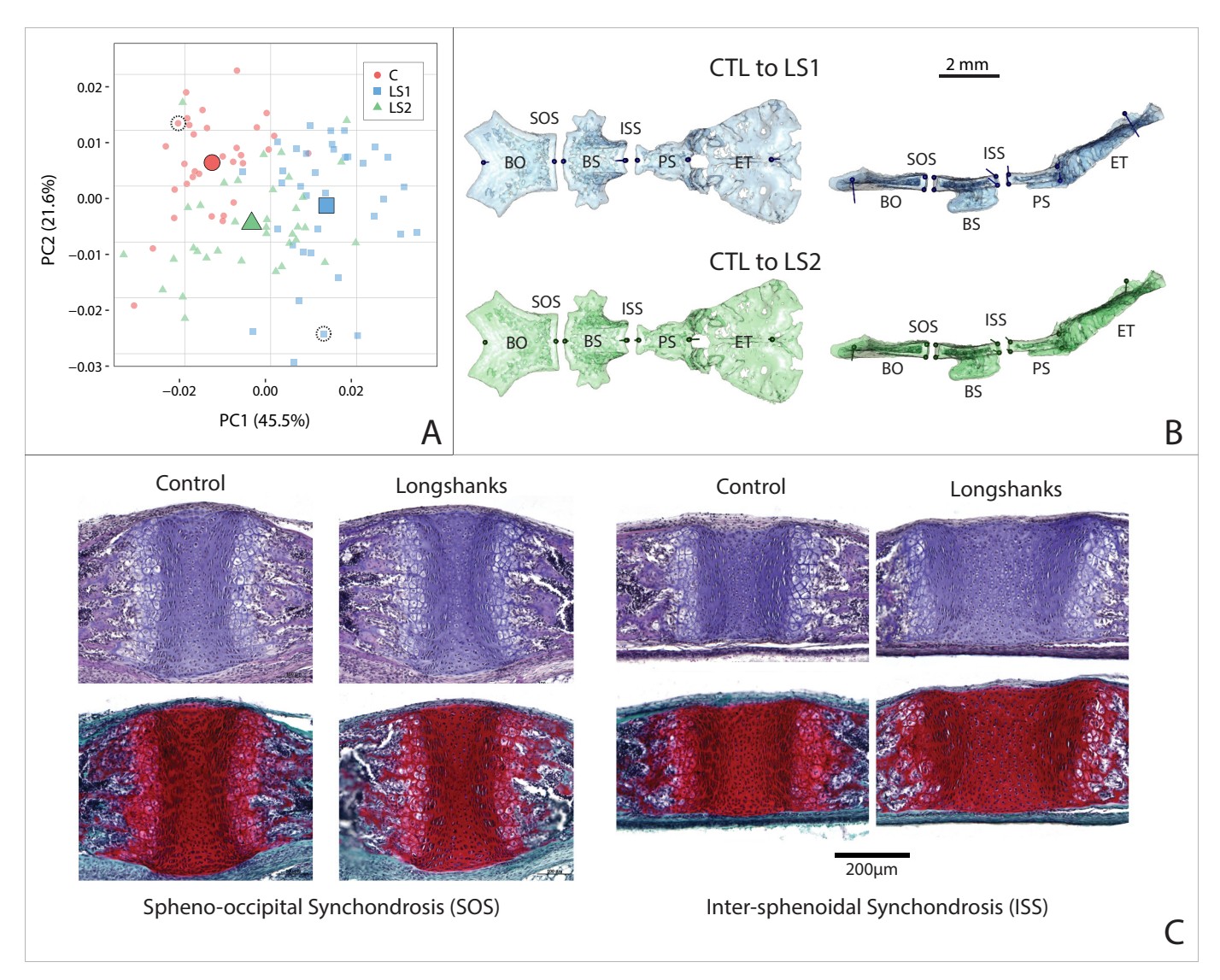

**Figure 6.** Cranial base contribution to the Longshanks phenotype. (**A**) Scatter plot of the first two PCs of litter size-adjusted Longshanks and Control cranial base Procrustes shape variables in generation 32 (F32) neonates. Large symbols indicate mean PC1 and PC2 scores for each respective cohort. (**B**) Neonate cranial bases with vectors of shape change at midline cranial base landmarks (magnified six times for visualization) showing shape transformations to go from the mean Control cranial base to the mean LS1 (blue) and mean LS2 (green) cranial base shapes. Views in (**B**) are dorsal (left) and lateral (right). Basi-occipital bone (BO), spheno-occipital synchondrosis (SOS), basi-sphenoid bone (BS), intersphenoidal synchondrosis (ISS), presphenoid bone (PS) and ethmoid (ET). (**C**) Sagittal midline histological sections stained in H + E (top) and safranin-o (bottom) showing differences in synchondrosis morphology of two extreme specimens in CTL and LS1 of approximately equal centroid size (indicated by dashed circles in A). The online version of this article includes the following source data and figure supplement(s) for figure 6:

**Source data 1.** Adult morphometric and landmark data.

**Figure supplement 1.** Neonate (P07) cranial base landmarks on the sagittal midline used in this study.

the cranium along the rostral caudal axis in Longshanks mice, independent of an overall increase in cranial size (*Figure 1*). In parallel, the cranium of adult Longshanks decreased in width between the zygomatic arches and reduced in vault height at the bregma and lambda (*Figure 1*). A similar cranium pattern was present in Longshanks neonates at P07, but with an ontogenically reduced craniofacial phenotype (*Figures 4* and *5*). LS2 mice appear to have a more subtle phenotype than LS1 in adulthood and at P07 (*Figures 1*, *3*, *5* and *6A*). This is not unexpected, as other genomic and

phenotypic differences in the response to selection between LS1 and LS2 have been documented previously (*Farooq et al., 2017*; *Castro et al., 2019*; *Cosman et al., 2019*).

The magnitudes of cranial shape change remain small in adult and neonate Longshanks relative to Controls, in comparison to stark morphological differences seen in skeletal mouse mutants (*Munroe et al., 2009*; *Gong, 2012*; *Holmes, 2012*). However, in this study, we are more interested in patterns of shape change rather than magnitudes of change. Over 20 generations, the main target of selection, the tibia, increased in length by just 15%. As such, we expected that secondary cranial shape changes, while potentially significant in terms of long-term evolution, would be subtle. Moreover, selection in the tibia appears to have increased the variation in cranium measures, such as centroid size and cranial shape (*Figure 3*). While the F20 LS1 and LS2 samples have new extreme cranium shapes not seen in earlier generations (*Video 2*), their effects on the mean shape are dampened substantially by the fact that many F20 mice still have crania that resemble F01 random-bred mice.

The net adult cranium phenotypic change in Longshanks manifested as a sequence of two distinct evolutionary shape changes. The cranium of both selected lines consistently reduced in vault height in the first 9 generations of tibia selection, and subsequently elongated and narrowed in the next 11 generations of selection (*Figure 3*). In comparison, the Control line shows minor cranial shape changes around the zygomatic and occipital regions, which are likely due to stochastic intergenerational variation in the Control line (i.e. drift), and/or to sampling artifacts, such as low sample size and family diversity in F09 Controls (*Table 1*). Similar evolutionary cranial shape changes have also been described in natural populations. Parmenter and colleagues noted that Gough Island mice, which differ significantly in body size compared to mainland relatives, have crania that are longer and narrower without differences in vault height, similar to the intergenerational changes in cranial shape between generations F09 and F20 in Longshanks (*Parmenter et al., 2016*). The mechanisms underlying these stepwise changes are unknown, in part due to a lack of F09 developmental data. However, the transition in cranium response could be related to temporal differences in allele frequency shifts up to, and subsequent to, generation F09, and/or to the nature in which cranial bones physically interact (e.g. the degree of perturbation that must accumulate in one cranium region before adjacent structures are significantly affected).

Importantly, despite shifts in the population means of tibia length and cranium shape over time (*Figure 2*), the cranium shape changes in adult LS1 and LS2 are uncorrelated with tibia length within each generation. In other words, selected individuals with relatively longer tibiae did not necessarily have derived cranial shapes. This suggests that residual cranial shape did not increase the chances of being selected as a breeder in our experiment, and therefore the observed change in residual cranial shape over time in Longshanks is most likely a correlated evolutionary response to tibia selection, mediated by underlying genetic correlations due to pleiotropy and/or linkage disequilibrium (*Lande, 1984*).

Correlated evolutionary responses have been documented in experimental populations primarily in terms of physiological and behavioral processes, such as starvation response, developmental

**Table 1.** Longshanks adult (F01, F09, F20) and neonate (F32) sample composition.

Target sample sizes of n = 40/group represent a compromise between sampling effort (e.g. scanning capacity) and the ability to detect small-to-moderate effect sizes at a power of 0.8 in the case of univariate analyses (Cohen's d = 0.25). For multivariate analyses (e.g. principal components analyses), target sample sizes of n = 40 with 50–68 landmarks produce highly repeatable covariance matrices (average RV coefficient of a sample's Procrustes-adjusted covariance matrix with 1000 covariance matrices derived from bootstrapped data > 0.99).

**Longshanks samples (n)**

|  | CTL | LS1 | LS2 |
| --- | --- | --- | --- |
| Generation 1 (F01) | 24 | 40 | 40 |
| Generation 9 (F09) | 23 | 40 | 40 |
| Generation 20 (F20) | 40 | 40 | 40 |
| Generation 32 (F32) | 32 | 36 | 36 |

duration, and voluntary wheel running (*Carter et al., 2000*; *Gammie et al., 2003*; *Davidowitz et al., 2016*; *Hardy et al., 2018*). Correlated responses in morphological phenotypes have also been characterized, but they have typically involved correlated change in the same anatomical structure, or indirect responses to body mass selection (*Tobler and Nijhout, 2010*). For example, selection for changes in wing spot pattern on the dorsal aspect of butterfly wings produced changes in ventral wing patterns (*Beldade et al., 2002*). To the best of our knowledge, our study presents the first strong case of correlated evolution of functionally unrelated anatomical structures with no direct musculotendinous or bony connections, in the mammalian skeletal system.

## Developmental correlates of shape change

From a developmental perspective, our investigation of Longshanks at one-week post partum revealed that Longshanks neonates have relatively flatter cranial bases, which develop by endochondral ossification, compared to Controls (*Figure 6*). Moreover, 2D geometric morphometrics at the midline and histology demonstrated that Longshanks neonates have a larger intersphenoidal synchondrosis with larger resting and proliferative zones, especially in its dorsal aspect (*Figure 6B,C*), much like differences observed in the tibial proximal epiphysis of 2-week-old Longshanks (*Marchini and Rolian, 2018*). In contrast, the spheno-occipital synchondrosis did not show any substantive size or shape differences among groups. Taken together, these results suggest that the intersphenoidal synchondrosis responded to selection on the tibia independently of the spheno-occipital synchondrosis and may have contributed to the Longshanks cranial phenotype.

It is important to note that our ontogenetic analysis captured a single developmental stage, hence the apparent uncoupling of the synchondroses may be because the spheno-occipital synchondrosis has developmentally important differences between Longshanks and Controls, but at a different time in development. For example, the spheno-occipital synchondrosis mineralizes faster than the intersphenoidal synchondrosis after P10 in CD-1 mice (*Wealthall and Herring, 2006*). Regardless, differential timing of spheno-occipital synchondrosis and intersphenoidal synchondrosis fusion are widespread across clades, suggesting these growth centers are partly under independent control. In humans, the intersphenoidal synchondrosis begins fusing at birth, whereas the spheno-occipital synchondrosis does not fuse until adolescence (*Madeline and Elster, 1995*). In domestic dogs, premature fusion of the spheno-occipital synchondrosis is a prominent feature of brachycephalic dogs (*Schmidt et al., 2013*). The partial developmental independence of the synchondroses is supported by knock-out studies of skeletogenic factors in rodents which have noted concerted changes in the postcranial epiphyses and synchondroses, but with differential responses in the respective synchondroses (reviewed in [*Vora, 2017*]). For example, in mice, Indian hedgehog (Ihh$^{-/-}$) knockout results in significantly more ectopic hypertrophic chondrocytes in the intersphenoidal synchondrosis than in the spheno-occipital synchondrosis (*Young et al., 2006*). The intrinsic differences between the two synchondroses may ultimately be associated with their distinct embryonic origins: the pre-sphenoid and basi-sphenoid bones originate from neural crest cells that commit to endochondral ossification, whereas the basi-occiptal bone forms from mesenchymal condensations of prechordal mesoderm (*McBratney-Owen et al., 2008*; *Richtsmeier and Flaherty, 2013*).

The exact ontogenetic mechanism by which the observed cranial base changes may lead to shape change across the entire cranium in Longshanks remains to be elucidated. There is extensive evidence for strong phenotypic covariation among cranial traits (i.e. morphological integration) causing indirect change throughout the cranium (*Bookstein et al., 2003*; *Goswami, 2006*; *Goswami et al., 2012*; *Singh et al., 2012*; *Bastir and Rosas, 2016*; *Neaux et al., 2019*). For example, direct basicranium perturbations by genetic mutations in mice generated predictable shape changes throughout the cranium (*Parsons et al., 2015*). Basicranium undergrowth by deletion of *Trsp* produces shortened faces and tall, domed calvaria. Conversely, cranial base overgrowth models, such as the *Pten*$^{-/-}$ mouse, resulted in flattened calvaria, elongated faces and reduced cranial width (*Ford-Hutchinson et al., 2007*; *Lieberman et al., 2008*; *Parsons et al., 2015*). Moreover, analysis of mutant mouse models have demonstrated that cranial base length and angle are interrelated with brain and face size (*Ross and Ravosa, 1993*; *Lieberman et al., 2008*). Notably, these transgenic models of basicranium perturbation also have concomitant tibia length changes (*Downey et al., 2009*; *Guntur et al., 2011*). While our study identified subtle evolutionary changes that occurred over time, rather than instant changes due to wholesale loss-of-function mutations, our findings are in general agreement with these cranium integration studies, suggesting that a common genetic

architecture can produce shared short-term developmental perturbations *and* long-term evolutionary change in anatomical structures that have no direct musculotendinous or skeletal links.

## Potential genetic basis of shape change

Our data suggest that, at the tissue level, correlated changes in tibia length and cranial shape are associated with similar changes in growth plate morphology. At this time, the genomic basis for these correlated skeletal changes remains unknown. The vertebrate tibia and cranium are complex structures, each with a highly polygenic architecture, thus finding candidate genes that caused changes in one or the other structure, let alone correlated changes in both, is not trivial. A previous study revealed that the genomic response to tibia selection at generation F17 in Longshanks involved substantive contributions from several loci of large effect against an infinitesimal polygenic background (*Castro et al., 2019*). By F17, LS1, and LS2 showed strong and parallel allele frequency changes in non-coding DNA in two large genomic regions spanning several topology associated domains (TADs) and containing putative cis-regulatory elements (*Castro et al., 2019*). Several genes in or near these loci, such as *Nkx3-2* and *Chst11*, have known cranium and limb pleiotropic effects in knock-out mice and human diseases (*Lettice et al., 1999*; *Tribioli and Lufkin, 1999*; *Akazawa et al., 2000*; *Klüppel et al., 2005*; *Provot et al., 2006*; *Chopra et al., 2015*). However, it is not clear whether these contributed to the overall F20 correlated cranium response, especially since allelic changes in non-coding regions are predicted to have smaller pleiotropic effects, if any, when compared to genomic changes in coding regions, which were rare at F17 (*Carroll, 2008*; *Stern and Orgogozo, 2008*; *Rice and Rebeiz, 2019*). Moreover, when cross-referenced against annotated mouse knockout phenotypes (Mouse Genome Informatics [MGI], http://www.informatics.jax.org), these genomic regions contain genes with singular effects on the cranium skeleton (e.g. *Tapt1 and Nfic*), hence it is also possible that the correlated response was driven in part by genetic linkage between limb- and cranium-specific genes swept along by selection. Interestingly, our query of MGI also shows that over 15 of the genes near the altered F17 loci influence brain development and morphology, and therefore could indirectly influence the F20 net cranium shape due to brain-skull physical interactions. More work will be necessary to shed light on the underlying genetic correlates of the Longshanks cranium phenotype, especially functional genomic and molecular analyses of the cranial base at multiple developmental stages.

The Longshanks experiment provides a robust example of correlated evolutionary changes in the vertebrate skeleton. These findings are made possible in part because the experimental design allowed us to assess skeletal change in a controlled environment, and without confounding variables (e.g. fluctuations in climate and/or food availability, presence of predators). At the same time, laboratory conditions in which a single selection pressure can be applied to the skeleton likely differ from the more complex multivariate selective pressures operating in nature. The simplified environment and strong directional selection pressures in experimental evolution studies may target alleles with detrimental pleiotropic effects more frequently than in natural settings (*Kawecki et al., 2012*). For example, in natural populations, correlated responses in the cranium to selection acting on the limb skeleton could be buffered by counteracting selection pressures, related to diet, olfaction, hearing and/or mate choice (*Samuels, 2009*; *Cox et al., 2012*; *Maestri et al., 2016*). This simplified evolutionary process is a strength of selection experiments. Indeed, by applying strong directional selection on a single trait, selection experiments can 'overwhelm' complex multivariate responses to selection operating in nature, including stabilizing selection on other structures, and in doing so reveal previously unknown pleiotropic developmental mechanisms that have the potential to drive correlated evolution among distinct anatomical structures.

## Conclusions

In this study, we characterized secondary skeletal responses to tibia selection that likely arose due to shared underlying developmental mechanisms between the cranium and tibia, specifically endochondral ossification. The limb and cranium are often considered separate modules in morphological analyses (*Young and Hallgrímsson, 2005*). Our results highlight the importance of considering evolution of the vertebrate skeleton in its entirety. Our study shows how indirect, and potentially non-adaptive, skeletal changes can occur due to developmental overlap among physically and functionally distant body parts. These findings have implications for how we reconstruct skeletal evolutionary

histories of extant and extinct mammalian lineages by providing empirical evidence that skeletal traits may arise solely as side effects of selection acting elsewhere.

# Materials and methods

## Key resources table

| Reagent type (species) or resource | Designation | Source or reference | Identifiers | Additional information |
|---|---|---|---|---|
| Mouse (*Mus musculus*) | Longshanks 1 | Campbell Rolian | LS1 | Stock: HSD:ICR |
| Mouse (*Mus musculus*) | Longshanks 2 | Campbell Rolian | LS2 | Stock: HSD:ICR |
| Mouse (*Mus musculus*) | Control | Campbell Rolian | CTL | Stock: HSD:ICR |
| Software, algorithm | *R, R packages geomorph, Morpho, RRPP* | *R Development Core Team, 2020*, *Schlager, 2020*, *Collyer and Adams, 2018*, *Adams et al., 2020* | n/a | |
| Software, algorithm | Amira | Visage Imaging, Berlin, Germany | Version 5.4.2 | |
| Software, algorithm | Python and Bash (Medical Imaging NetCDF library) | *Percival et al., 2019*, *Devine et al., 2020* | Python 3.6 Bash-5.1 MINC 2.0 | https://github.com/BIC-MNI/minc-toolkit-v2 https://github.com/jaydevine/Landmarking |
| Other | Haematoxylin (for Weigert's) | Sigma | H3136 | Histological stain |
| Other | Iron (III) chloride (for Weigert's) | Sigma | 157740 | Histological stain |
| Other | Fast Green (FCF) | Sigma | F7252 | Histological stain |
| Other | Safranin-O | Sigma | S2255 | Histological stain |
| Other | Gill's Haematoxylin | Sigma | GHS332 | Histological stain |
| Other | Eosin Y | Sigma | 588X | Histological stain |

## Animal samples

All animal procedures were approved by the Health Sciences Animal Care Committee at the University of Calgary (AC13-0077) and (AC17-0026) and performed in accordance with best practices outlined by the Canadian Council on Animal Care. For more information on the husbandry methods and selective Longshanks breeding regimen, see *Marchini et al., 2014*.

We collected 8-week-old, non-breeder Longshanks mice (N = 327) from generations 1, 9, and 20 across three experimental lines: Longshanks 1 (hereafter LS1), Longshanks 2 (LS2), and Control (CTL) to study changes in adult cranial shape at the beginning, middle, and end of the selection process, respectively (for group sample sizes, see *Table 1*). Each group was as sex and family balanced as possible to account for differences due to sexual dimorphism and/or family diversity (*Karp et al., 2017*). To investigate the developmental basis of the Longshanks cranium, we generated postnatal day seven (P07) neonates (N = 104) from F31 Longshanks mice (*Table 1*). The three lines have not actively undergone artificial selection since generation F22 and are maintained as experimental populations. We selected P07 as our developmental time point as this is when Longshanks tibiae are growing fastest, and the cranial skeleton is still actively growing (*Vora et al., 2015*; *Farooq et al., 2017*; *Marchini and Rolian, 2018*).

## X-ray micro-computed tomography (μCT)

We performed X-ray micro-computed tomography (μCT). We used a Skyscan 1173 v1.6 μCT scanner (Bruker, Kontich, Belgium) to acquire whole-body scans of the adults and separate scans of the neonate cranium and tibiae. We obtained adult samples from frozen archived carcasses at each generation, while F32 neonates were scanned the day they were euthanized. In addition, we scanned the corresponding right hindlimb of each neonate that underwent cranium scanning. Adult scans were acquired at 70–80 kV and 60–75 μA with 44.73 μm isotropic voxels and no filter, while neonates

were scanned at a resolution of 17.04 µm isotropic voxels with otherwise identical parameters. Stack reconstructions were performed using NRecon v1.7.4.2 (Bruker, Kontich, Belgium).

## Histology

We dissected neonate crania after scanning them and fixed them in 10% neutral buffered formalin (NBF) (Thermo Scientific) for 48 hr, with NBF replacement every 24 hr. Fixed cranium tissues were then transferred to a decalcifying solution (Cal-Ex II, Fisher Chemical) for 72 hr with daily solution changes. After decalcification, a rectangular portion of the cranial base containing both basicranial synchondroses was dehydrated, embedded it in paraffin, and sectioned in the sagittal plane at 12 µm. Sections were deparaffinized in Slide Brite (Jones Scientific Products, Inc) and subsequently stained. The slides of a specimen were stained in an alternating fashion with two stains: (1) Wiegert's Iron Haematoxylin (Sigma), 0.05% Fast-Green (FCF) (Sigma), counterstained in 0.1% Safranin-o solution (Sigma); or (2) Gill's Haematoxylin #3 (Sigma), rinsed in 70% ethanol, and counterstained with 1% alcoholic Eosin Y (Sigma). We imaged sagittal midline sections using an Axio Scan.Z1 slide scanner (Ziess, Oberkochen, Germany) at ×10 magnification and qualitatively evaluated differences in growth plate size and morphology.

## Landmarking

µCT adult and neonate crania scans were subjected to a novel image registration-based pipeline to automatically detect landmarks for a geometric morphometrics shape analysis (*Percival et al., 2019*). Automated landmarking improves data standardization and can be used to quickly process very large sample sizes while reducing intraobserver errors, such as landmark placement drift (*Fruciano, 2016*; *Devine et al., 2020*). Automated landmarking involves volumetric registration using a global affine alignment of the skull volumes, followed by a dense non-linear deformation between each cranium and a reference atlas. Here, the atlas is an average volume, with a standardized landmark configuration, that best minimizes intensity differences from the rest of the sample. We used 68 3D landmarks for the adults (*Figure 1—figure supplement 4*, *Supplementary fle 4*) and 50 3D landmarks for the neonates (*Figure 4—figure supplement 2*, *Supplementary file 5*).

We computed the affine transformations with a multi-resolution framework, where the µCT volumes are translated, scaled, rotated, and sheared at progressively higher resolutions until their affine alignment with the atlas is maximized (*Lerch et al., 2010*). We computed the non-linear transformations with the multi-resolution SyN (Symmetric Normalization) algorithm (*Avants et al., 2011*), which involves symmetrically flowing an image pair into one another along a velocity field. We then recovered, concatenated, and inverted the transformations, and finally propagated the atlas landmarks along this path to the original image space for analysis. All image processing was performed with the open-source MINC (Medical Imaging NetCDF) toolkit (https://github.com/BIC-MNI/minc-toolkit-v2).

In addition to investigating overall neonate cranium shape, we characterized cranial base shape with two-dimensional (2D) landmarks at the sagittal midline. We used a 12 landmark set highlighting the vertices of the developing basicranial bones which provides information about the shape of the sagittal cross-section of the basicranial synchondroses (*Figure 6—figure supplement 1*; *Supplementary file 6*). Landmarks at the midline were placed in Amira v.5.4.2 (Visage Imaging, Berlin, Germany) by one observer (CMU) blind to the identity of the specimens. Adult tibiae lengths were quantified in Amira by calculating the distance, in mm, between two landmarks that we placed on the distal tip of the lateral malleolus and most lateral point on the proximal epiphysis, two anatomical points that were demonstrated to have high homology and repeatability (*Cosman et al., 2016*). Because neonate tibia length is not fully visible in the scans due to small or absent secondary ossification centers (*Moss, 1977*), neonate tibia measurements were obtained from the distance, in mm, between landmarks placed on the distal and proximal ends of the ossified tibial diaphysis on the rostral edge along the sagittal midline of the tibia.

## Geometric morphometrics

Analyses were performed on the R/Rstudio computational platform (*R Development Core Team, 2020*). We investigated shape cranial differences by superimposing the adult and neonate landmark configurations into age-specific morphospaces via Generalized Procrustes Analysis. To study the

influence of selection on cranial shape, we first corrected for confounding variables known to alter adult and neonatal morphology.

In the adult sample, we controlled for the effects of sex and size. Upon regressing shape on sex, we observed that sex accounted for a small but significant amount of variation (2.2%), although there were no sex-specific differences in cranial responses to selection (data not shown). Using sex-adjusted residuals, we investigated allometry in the Longshanks cranium to parse out how much of the cranial selection response, if any, could be attributed to changes in skeletal size. While Procrustes superimposition removes scale, it does not account for differences in biological shape that are associated with size that is allometry (*Klingenberg, 2016*). Because Longshanks mice are skeletally larger in relation to body mass in the post-cranium (*Sparrow et al., 2017*), we employed a pooled within-group analysis of covariance (ANCOVA) of cranium centroid size on body mass to determine whether the same trend exists in the Longshanks cranium. Mean centroid size after accounting for body mass was significantly different among lines at F20 (F = 14.97, p<0.001), with Longshanks LS1 and LS2 lines having larger crania than Controls (*Figure 1—figure supplement 2*) (Tukey's post-hoc test, LS1vsLS2 p=0.947, LS1vsCTL p<0.001, LS2vsCTL p<0.001). There was no difference in mean centroid size, after controlling for covariation with body mass, among founder (F01) samples (*Figure 1—figure supplement 2*) (LS1vsLS2 p=0.896, LS1vsCTL p=0.999, LS2vsCTL p=0.999).

For the neonate sample, we controlled for the effects of litter size but not sex, due to uncertainties in assigning sex anatomically in neonates. After regressing cranial size and tibia length on litter size, we observed a strong negative correlation (Pearson, $r = -0.72$, p<0.001). LS2, which had litter sizes that were ~ 2 pups larger than LS1 and ~ 4 pups larger than controls on average, exhibited significantly smaller centroid sizes than LS1 and Controls (ANOVA, F = 19.51, Tukey's HSD, LS1vsLS2 p<0.001, LS1vsCTL = 0.764, LS2vsCTL p<0.001) (*Figure 1—figure supplement 1*, *Supplementary file 1*). Thus, we performed our neonate analyses with Procrustes shape variables and univariate measurements, such as tibia length, adjusted for litter size effects.

Group differences in adult and neonate cranial morphology were evaluated using principal component analyses. We assessed whether group mean shapes, independent of size and/or sex, were statistically significantly different using a randomized residual (1000 permutations) Procrustes ANCOVA (*Goodall, 1991*; *Collyer et al., 2015*). Post-hoc pairwise tests compared differences in least-squares means between groups (*Collyer and Adams, 2018*). For visualizations of cranial shape differences between lines, we used deformation heatmaps and cranial meshes with vectors of shape change that depict transformations between group means. All geometric morphometric analyses were performed in R with the *geomorph*, *Morpho,* and *RRPP* packages (*Schlager, 2017*; *Schlager, 2020*; *Collyer and Adams, 2018*; *Adams et al., 2020*).

## Acknowledgements

The authors are grateful to the Animal Resource Center staff at the University of Calgary for the continued care they provide to the Longshanks colonies. The authors are also indebted to Jason Anderson and Jessica Theodor for granting access to the SkyScan 1173 uCT scanner for data acquisition. The authors are also grateful to Dragana Ponjevic for expertise in histology and Heather Jamniczky for guidance in data analysis. This work was supported by a Natural Sciences and Engineering Research Council Canada Graduate Scholarships – Master's award to CMU, a Natural Sciences and Engineering Research Council Discovery Grant 4181932 to CR, the McCaig Institute for Bone and Joint Health, and the Faculty of Veterinary Medicine at the University of Calgary (http://www.vet.ucalgary.ca).

## Additional information

### Funding

| Funder | Grant reference number | Author |
| --- | --- | --- |
| Natural Sciences and Engineering Research Council of Canada | Discovery Grant 4181932 | Campbell Rolian |
| University of Calgary | Faculty of Veterinary | Campbell Rolian |

| | Medicine | |
| --- | --- | --- |
| Natural Sciences and Engineering Research Council of Canada | Canada Graduate Scholarship - Masters | Colton Michael Unger |

The funders had no role in study design, data collection and interpretation, or the decision to submit the work for publication.

### Author contributions

Colton M Unger, Conceptualization, Data curation, Formal analysis, Validation, Investigation, Visualization, Methodology, Writing - original draft, Writing - review and editing; Jay Devine, Resources, Software, Formal analysis, Validation, Investigation, Visualization, Methodology, Writing - review and editing; Benedikt Hallgrímsson, Resources, Software, Formal analysis, Methodology, Writing - original draft, Writing - review and editing; Campbell Rolian, Conceptualization, Resources, Data curation, Software, Formal analysis, Supervision, Funding acquisition, Validation, Investigation, Visualization, Methodology, Writing - original draft, Project administration, Writing - review and editing

### Author ORCIDs

Colton M Unger (ID) https://orcid.org/0000-0001-6793-8173
Jay Devine (ID) https://orcid.org/0000-0003-0600-1058
Benedikt Hallgrímsson (ID) https://orcid.org/0000-0002-7192-9103
Campbell Rolian (ID) https://orcid.org/0000-0002-7242-342X

### Ethics

Animal experimentation: All animal procedures were approved by the Health Sciences Animal Care Committee at the University of Calgary (Protocols AC13-0077 and AC17-0026) and performed in accordance with best practices outlined by the Canadian Council on Animal Care.

### Decision letter and Author response

Decision letter https://doi.org/10.7554/eLife.67612.sa1
Author response https://doi.org/10.7554/eLife.67612.sa2

## Additional files

### Supplementary files

• Supplementary file 1. Morphometric data for adult mice among lines and generations. Body mass data represent means and SEM, whereas centroid size and tibia length are least squared means and SEM. Superscripts denote significant differences in means ($p<0.05$) between a given group and: Controls [CTL], Longshanks Line 1 [LS1], Longshanks Line 2 [LS2], from either: Generation 1 [F01], Generation 9 [F09], or Generation 20 [F20], as determined using Tukey's HSD tests. Differences in body mass were determined by ANOVA, whereas tibia length and centroid size differences were assessed by ANCOVA with body mass as a covariate. Bold and italic superscripts indicate significant intergenerational differences and intragenerational differences, respectively.

• Supplementary file 2. Mean Euclidean distances. Euclidean distances between the multivariate mean PC scores of each group, based on Procrustes shape data adjusted for sex only (above diagonal), or sex and cranial centroid size (below diagonal). The only non-significant Euclidean distance, based on a post-hoc Procrustes ANCOVA, is indicated in bold.

• Supplementary file 3. Morphometric data for neonate mice among lines and generations. Tibia and centroid size data represent least squared means (SEM) and litter sizes are means (SEM). Differences in litter size were determined by ANOVA, whereas tibia length and centroid size differences were assessed by ANCOVA with litter size as a covariate. Superscripts denote significant differences in means ($p<0.05$) between a given group and: Controls [CTL], Longshanks Line 1 [LS1], Longshanks Line 2 [LS2].

- Supplementary file 4. Adult cranium landmarks and their anatomical definitions.
- Supplementary file 5. Neonate cranium landmarks and their anatomical definitions.
- Supplementary file 6. Neonate cranial base landmarks and their anatomical definitions.
- Transparent reporting form

## Data availability

All data generated or analysed during this study are included in the manuscript and supporting source data files.

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
