## [Decision Letter]

**Acceptance summary:**

The authors report skull shape changes in mice artificially selected for a phenotype in a different part of the body, specifically limb length. This result is of high importance for both evolutionary biologists and developmental biologists. That is, interpretations of assumed isolated phenotypic modules should proceed with caution because they may actually be unexpectedly connected based on shared underlying mechanisms, the developmental genetic basis of which will be of interest to expansively characterize.

**Decision letter after peer review:**

[Editors’ note: the authors submitted for reconsideration following the decision after peer review. What follows is the decision letter after the first round of review.]

Thank you for submitting your work entitled "Selection for increased tibia length in mice alters skull shape through parallel changes in developmental mechanisms" for consideration by *eLife*. Your article has been reviewed by 2 peer reviewers, and the evaluation has been overseen by a Reviewing Editor and a Senior Editor. The following individual involved in review of your submission has agreed to reveal their identity: Carlos Infante (Reviewer #1).

Our decision has been reached after consultation between the reviewers. Based on these discussions and the individual reviews below, we regret to inform you that your work will not be considered further for publication in *eLife*.

The reviewers agree that the manuscript presents a robust and interesting morphometric dataset. However, both reviewers also agree that the data are over-interpreted and that the primary conclusions of the manuscript are not well supported. Unfortunately, a significant amount of additional research seems to be required to establish the genetic or molecular mechanisms underlying the coordinated changes in limb and skull morphology; more research than could be completed in the normal time frame for *eLife* revisions. Therefore, the conclusion reached by the reviewers and the editors in discussion is that the manuscript is not appropriate for *eLife* in its current format.

*Reviewer #1:*

This is a well-written and detailed manuscript that I recommend for publication in *eLife*. The authors describe correlated changes between limb length and skull shape in a line of laboratory mice that have undergone generations of artificial selection for limb length (Longshanks mice) using both 2D and 3D morphometric analyses. The key findings are the demonstration of unexpected effects on non-target tissues (skull shape) of this artificial selection, and the identification of cellular changes in the off-target tissue underlying the effect. The discussion cautions that the interpretation of morphological evolution of the vertebrate skeleton in terms of distinct, isolated modules may ignore unexpected connections based on underlying genetic and developmental mechanisms.

1. The study relies on morphometric analyses and some histology on the SOS and ISS in neonate crania to establish a mechanism for the correlated changes in the shape of the ISS. This argument would be strengthened with some additional data on what is happening at the cellular level of the ISS and the growth plate of the tibia to support that the same molecular mechanism is being deployed to produce the changes in cellular organization seen in the ISS. But I understand that these are beyond the scope of this paper.

2. I have a concern about the interpretation of the results in the broader context of the evolution of skeletal adaptations. I agree that unexpected correlated changes could result from the selection on a particular feature, but I also could imagine that other selective pressures could prevent could prevent correlated changes. A laboratory strain of mice may not have the full selective environment of a natural population, and perhaps changes in limb length would proceed through alternative developmental and genetic mechanisms. Maybe the associated limb/cranium changes are a path of least resistance only available in a laboratory strain?

*Reviewer #2:*

The manuscript by Unger et al., identifies correlated evolution of the skull and tibia in Longshanks mice. The authors find that the response to artificial selection on tibia length directly or indirectly facilitated the evolution of skull shape in two separate selection lines. Most interesting, although both lines experienced different patterns of allelic response and tibial evolution, they both depict similar patterns of skull shape change over the course of 20 generations of selection. They find that in the first 9 generations, cranial vault height is changing. From generation 10 and onward, cranial width is reduced in parallel with elongation of the snout. The authors come to the conclusion that correlated evolution of the skull and tibia may be due to pleiotropy and discuss potential candidates.

This is a well written and easy to read manuscript, with a robust and interesting morphometric dataset. I particularly like the use of heat maps overlaid on the skull to depict shape change. These figures are far more readable than typical splines, warps, and vector diagrams used in most morphometric studies. In all, the morphological data presented produce a convincing argument for indirect correlated evolution via pleiotropy.

My concerns largely deal with the figures and interpretations presented in the discussion.

1. After some fine tuning, I think that Supplemental Figure 8 should be Figure 1 in the manuscript and not hidden away in the supplement. Correlated trait evolution is the crux of the argument of this manuscript, and this figure depicts that quite elegantly. Sometimes the most basic figures are best at getting the message across. To make the visual more compelling, I suggest making each plot specific to LS1, LS2, and Ctl. In each panel, plot generation F01, F09, and F20 so readers can see the gradual shifting to the upper right corner of the plot as skull size increases with tibia size in LS1/2. This shift will be noticeably absent in Ctl.

2. My greatest concern is that the authors mechanistically attribute major loci associated with tibia length evolution not just to the evolution of the skull, but also the shift in the skull response to selection that occurs between generations 9 and 10 (lines 403-412). For example, previous studies on the Longshank mice by (Castro et al., 2019) looked at genome-wide differences between F0 and F17, in which they identified NKX-3.2 as a major effect locus that had almost swept to fixation in both LS1 and LS2 at generation 17. The authors note that "complete ablation of nkx-3.2 results in cranial base truncation and premature synchondrosis fusion in mice." From this they suggest that selection for nkx-3.2 downregulation in the tibia may have caused indirect cranial changes in the skull via pleiotropic effects. In lines 413-421 the authors suggest the switch in the response of the skull may reflect selection on genes affecting all cranial growth plates and their response to systemic factors.

This is an over-interpretation of the correlated evolution. Although the manuscript demonstrates a correlation between tibia length and skull size/shape variation, this does not mean that a major contributor to tibia evolution will also be a major contributor to cranial evolution. Both phenotypes are likely polygenic traits, and any subset of genes responsible for tibial elongation might disproportionately contribute to skull size/shape. More importantly, the genomic dataset is the limiting factor in this interpretation. Only two generations were sampled in the (Castro et al., 2019) study, F0 and F17. We don't know what the pattern of allelic response between F0-F9 looked like. It is certainly possible that nkx-3.2 started to sweep towards fixation at F9 explaining the shift in skull shape, but without the generational data this is impossible to know. It is difficult to therefore extend any findings in the study by (Castro et al., 2019) to the current dataset without generating a more complete genomic dataset as has been done in other artificial selection studies (Burke et al., 2014; Hardy et al., 2017; McKenna, 2020). I am not suggesting this be done for this manuscript. I think this is a powerful stand-alone morphometric study. I suggest reworking this section of the discussion to more broadly discuss the potential similarities and differences that might exist between the skull and tibia: does genetic modularity influence variational states of the skull and tibia, thereby influencing which genes responding to selection on tibia length also impact skull size/shape? It's appropriate to use nkx3.2 as a toned down example. Do bones within the skull have different variational states and can this explain the shift in response to selection, e.g. one bone responds early and another responds late?

3. My last concern lies within the Discussion section (470-491) and includes Figure 6. Although I agree that change in the size of any bone in the skull may exert physical effects on all other bones, the data for this are simply not present in the study making its discussion all conjecture. I think this would make for a fabulous follow-up experimental study, but it is not currently supported by the data in the manuscript. I suggest replacing this section with a discussion of the novelty of this manuscript: that is, a direct observation of correlated evolution of morphological structures. Scouring the literature of experimental evolution studies, one can find numerous examples of correlated evolution of physiological processes (Burke et al., 2014; Davidowitz et al., 2016; Hardy et al., 2017; McKenna, 2020), but few have demonstrated correlated evolution of anatomical structures that require integration of different cellular parameters (excluding butterfly eyespots that are on the same appendage (Allen et al., 2008; Beldade et al., 2002) or allometric relations of an appendage to overall body size (Frankino et al., 2019; Tobler and Nijhout, 2010)). I think such a discussion would illustrate the utility of the Longshanks experiment in its ability to identify how skeletal elements are semi-discrete: some genes are pleiotropic and others are modularly tuned, and from this we get the amazing malleability of the skeleton that has produced the size and shape variation present in mammals.

[Editors’ note: further revisions were suggested prior to acceptance, as described below.]

Thank you for submitting your article "Selection for increased tibia length in mice alters skull shape through parallel changes in developmental mechanisms" for consideration by *eLife*. Your article has been reviewed by 2 peer reviewers, and the evaluation has been overseen by George Perry as the Senior and Reviewing Editor. The following individual involved in review of your submission has agreed to reveal their identity: Mark Grabowski (Reviewer #1).

Summary:

The authors report skull shape changes in mice artificially selected for a phenotype in a different part of the body, specifically limb length. This result is of high importance for both evolutionary biologists and developmental biologists. That is, interpretations of assumed isolated phenotypic modules should proceed with caution because they may actually be unexpectedly connected based on shared underlying mechanisms, the developmental genetic basis of which will be of interest to expansively characterize.

Essential Revisions:

Your revision was effective. Both reviewers were supportive overall, with a relatively small number of clarifying revision requests indicated in their recommendations for authors, all of which we consider essential for you to make. In particular, I agree with the comment from reviewer 1 in terms of the overuse of acronyms/abbreviations; other than for extremely common and recognizable ones, e.g. SNP, readability is greatly improved by writing these out.

*Reviewer #1 :*

The authors tested whether selection for increased tibia length produced indirect responses in the cranium though shared endocranial ossification processes in experimental lines of mice. They compared 3D shape data of two selected mice lines adult crania to the control lines across three generations of the selection experiment, which ran for 20 generations. Neonates were also collected from the three lines, to test if the shape changes in the skull were visible in early ontogeny and investigate its developmental basis. The authors controlled for the effects of sex and size in adults and litter size in neonates. Shape changes were compared using Procrustes ANCOVA, and visualized using PCA and deformation heatmaps. Results showed that the selected mice had larger crania independent of body mass, and a different shape, and this process appears to have occurred in a step-wise fashion throughout the selection experiment. They also showed that these changes appeared in the neonates as well, suggesting that the changes seen in the adults were due to developmental shifts.

I found the manuscript easy to read and had very few comments, possibly because it has already been through review previously. The authors have done a good job addressing a complex topic and walking us through how and why each part of the analysis was done – the manuscript has a good flow. Nice looking figures too.

1. I think it would make sense for the authors to very briefly (a sentence or 2) address why they suggest why the basicranium and tibia are unrelated. Though I am not up on my mouse anatomy, I can easily see changes in for example tibia length, affecting muscle insertion points on the shoulder/neck area, which might also lead to correlated changes in muscles in this area then attached to the cranium – resulting in functional integration, of a sort. Reading the response to reviewer letter, this may have been taken out due to a reviewer concern, but I think something on this topic should be briefly included.

2. Endochondral ossification – the authors should clarify why they suggest that a similar process through which the basicranium and long-bones develop would result in correlated evolution. This may have been in an earlier draft of the manuscript that was previously reviewed and removed, but I am a new reviewer and this is the first version I have seen. Citations supporting their reasoning would of course be welcome, I just think this should be made clear in the introduction – around Lines 62/65. Some of this may be a case of simply moving some sentences from then discussion to the introduction.

---

## [Author Response]

[Editors’ note: the authors resubmitted a revised version of the paper for consideration. What follows is the authors’ response to the first round of review.]

Reviewer #1:1. The study relies on morphometric analyses and some histology on the SOS and ISS in neonate crania to establish a mechanism for the correlated changes in the shape of the ISS. This argument would be strengthened with some additional data on what is happening at the cellular level of the ISS and the growth plate of the tibia to support that the same molecular mechanism is being deployed to produce the changes in cellular organization seen in the ISS. But I understand that these are beyond the scope of this paper.2. I have a concern about the interpretation of the results in the broader context of the evolution of skeletal adaptations. I agree that unexpected correlated changes could result from the selection on a particular feature, but I also could imagine that other selective pressures could prevent could prevent correlated changes. A laboratory strain of mice may not have the full selective environment of a natural population, and perhaps changes in limb length would proceed through alternative developmental and genetic mechanisms. Maybe the associated limb/cranium changes are a path of least resistance only available in a laboratory strain?Reviewer #2:The manuscript by Unger et al., identifies correlated evolution of the skull and tibia in Longshanks mice. The authors find that the response to artificial selection on tibia length directly or indirectly facilitated the evolution of skull shape in two separate selection lines. Most interesting, although both lines experienced different patterns of allelic response and tibial evolution, they both depict similar patterns of skull shape change over the course of 20 generations of selection. They find that in the first 9 generations, cranial vault height is changing. From generation 10 and onward, cranial width is reduced in parallel with elongation of the snout. The authors come to the conclusion that correlated evolution of the skull and tibia may be due to pleiotropy and discuss potential candidates.This is a well written and easy to read manuscript, with a robust and interesting morphometric dataset. I particularly like the use of heat maps overlaid on the skull to depict shape change. These figures are far more readable than typical splines, warps, and vector diagrams used in most morphometric studies. In all, the morphological data presented produce a convincing argument for indirect correlated evolution via pleiotropy.My concerns largely deal with the figures and interpretations presented in the discussion.1. After some fine tuning, I think that Supplemental Figure 8 should be Figure 1 in the manuscript and not hidden away in the supplement. Correlated trait evolution is the crux of the argument of this manuscript, and this figure depicts that quite elegantly. Sometimes the most basic figures are best at getting the message across. To make the visual more compelling, I suggest making each plot specific to LS1, LS2, and Ctl. In each panel, plot generation F01, F09, and F20 so readers can see the gradual shifting to the upper right corner of the plot as skull size increases with tibia size in LS1/2. This shift will be noticeably absent in Ctl.2. My greatest concern is that the authors mechanistically attribute major loci associated with tibia length evolution not just to the evolution of the skull, but also the shift in the skull response to selection that occurs between generations 9 and 10 (lines 403-412). For example, previous studies on the Longshank mice by (Castro et al., 2019) looked at genome-wide differences between F0 and F17, in which they identified NKX-3.2 as a major effect locus that had almost swept to fixation in both LS1 and LS2 at generation 17. The authors note that "complete ablation of nkx-3.2 results in cranial base truncation and premature synchondrosis fusion in mice." From this they suggest that selection for nkx-3.2 downregulation in the tibia may have caused indirect cranial changes in the skull via pleiotropic effects. In lines 413-421 the authors suggest the switch in the response of the skull may reflect selection on genes affecting all cranial growth plates and their response to systemic factors.This is an over-interpretation of the correlated evolution. Although the manuscript demonstrates a correlation between tibia length and skull size/shape variation, this does not mean that a major contributor to tibia evolution will also be a major contributor to cranial evolution. Both phenotypes are likely polygenic traits, and any subset of genes responsible for tibial elongation might disproportionately contribute to skull size/shape. More importantly, the genomic dataset is the limiting factor in this interpretation. Only two generations were sampled in the (Castro et al., 2019) study, F0 and F17. We don't know what the pattern of allelic response between F0-F9 looked like. It is certainly possible that nkx-3.2 started to sweep towards fixation at F9 explaining the shift in skull shape, but without the generational data this is impossible to know. It is difficult to therefore extend any findings in the study by (Castro et al., 2019) to the current dataset without generating a more complete genomic dataset as has been done in other artificial selection studies (Burke et al., 2014; Hardy et al., 2017; McKenna, 2020). I am not suggesting this be done for this manuscript. I think this is a powerful stand-alone morphometric study. I suggest reworking this section of the discussion to more broadly discuss the potential similarities and differences that might exist between the skull and tibia: does genetic modularity influence variational states of the skull and tibia, thereby influencing which genes responding to selection on tibia length also impact skull size/shape? It's appropriate to use nkx3.2 as a toned down example. Do bones within the skull have different variational states and can this explain the shift in response to selection, e.g. one bone responds early and another responds late?3. My last concern lies within the Discussion section (470-491) and includes Figure 6. Although I agree that change in the size of any bone in the skull may exert physical effects on all other bones, the data for this are simply not present in the study making its discussion all conjecture. I think this would make for a fabulous follow-up experimental study, but it is not currently supported by the data in the manuscript. I suggest replacing this section with a discussion of the novelty of this manuscript: that is, a direct observation of correlated evolution of morphological structures. Scouring the literature of experimental evolution studies, one can find numerous examples of correlated evolution of physiological processes (Burke et al., 2014; Davidowitz et al., 2016; Hardy et al., 2017; McKenna, 2020), but few have demonstrated correlated evolution of anatomical structures that require integration of different cellular parameters (excluding butterfly eyespots that are on the same appendage (Allen et al., 2008; Beldade et al., 2002) or allometric relations of an appendage to overall body size (Frankino et al., 2019; Tobler and Nijhout, 2010)). I think such a discussion would illustrate the utility of the Longshanks experiment in its ability to identify how skeletal elements are semi-discrete: some genes are pleiotropic and others are modularly tuned, and from this we get the amazing malleability of the skeleton that has produced the size and shape variation present in mammals.

Through detailed analyses of cranial shape and development in a large, multigenerational dataset, we describe four important findings: (i) 20 generations of selection for increased tibia length significantly altered adult Longshanks cranial shape, making it flatter, narrower, and longer than in wildtype mice; (ii) Cranial shape changes occurred in a stepwise process, with cranial vault flattening appearing over the first 9 generations, and cranial elongation appearing between generations 9 and 20 of the selective breeding experiment; (iii) Similar cranial shape changes are present as early as seven days postnatal, indicating that changes in skull development likely underlie the shape changes seen in adult crania;, and (iv) Developmentally, the change in cranial shape in Longshanks is associated with changes to the cellular architecture in one of the two synchondroses of the cranial base, mirroring changes in the structure of the Longshanks tibia growth plate.

We drew two main conclusions from these morphometric and histological data: (1) strong selection for increased tibia length can lead to indirect selection responses in cranium shape, and (2) these correlated shape changes likely arise due to ontogenetic changes in both structures, specifically parallel changes in the cellular mechanisms of endochondral ossification that govern tibia and cranial base growth. The reviewers who assessed our manuscript were largely supportive, but, to paraphrase, felt that our discussion on the generalizability (Reviewer 1), on the genomic basis of the correlated change, and on the contribution to cranial shape changes of physical interactions during skull development (Reviewer 2), was too speculative. The consensus among editors and reviewers was that we over‐interpreted our morphometric and ontogenetic data, and that additional genetic/molecular data would be required to support our two main conclusions.

We agree that aspects of our discussion were overly speculative, and ultimately detracted from the evidence of our robust morphometric and ontogenetic data in support of our conclusions. Accordingly, we have overhauled our discussion to address the reviewers’ concerns. We have reorganized the discussion to highlight more clearly and sequentially the phenotypic, ontogenetic, and genomic implications of our study. First, we summarize and discuss the observed correlated phenotypic changes in Longshanks (lines 396‐456) and situate them within the broader literature on correlated evolution of morphological traits, as suggested by Reviewer 2 (comment #3). Next, we have consolidated our discussion of the developmental implications of our study, including aligning better the Longshanks phenotype with data from transgenic models (lines 486505), and removing the epigenetic model of physical interactions relating cranial base developmental differences to the Longshanks adult cranial phenotype, as requested by Reviewer 2 (comment #3).

Third, in response to Reviewer 2 (comment #2), we have consolidated and significantly revised our discussion of the potential genomic basis of the correlated change in a separate section entitled “Potential Genetic Basis of Shape Change” (lines 510‐536). We discuss what our previously published paper on the genomic response to selection in Longshanks (Castro et al., 2019) does, and does not, tell us about genomic correlates of shape change in the tibia and cranium. We highlight strong and parallel allele frequency changes in non‐coding DNA in two large genomic regions spanning several topology‐associated domains (TADs) and containing putative cis‐regulatory elements (lines 514‐523). We stress that, as non‐coding regions, these loci may not be responsible for the parallel change observed in the skull (lines 523‐526). In addition, using data from the Mouse Genome Informatics consortium (MGI), we highlight additional genes near these TADs that singularly affect skull and/or brain growth, thus providing potential alternative pathways underlying cranial shape changes, in this case more likely through linkage than through pleiotropy (lines 527‐534). We believe this new section provides more balanced speculation, informed by our previous genomic data and by MGI, yet without attributing the correlated changes to any one candidate gene or regulatory element.

Lastly, in response to Reviewer 1 (comment 2), we include a separate limitations section (lines 539‐560) that specifically addresses the generalizability of our findings. We stress that the conditions in our lab‐based selection experiment likely do not reflect the multivariate selection regimes that organisms are more likely subject to in nature. Hence, our findings, despite providing strong evidence of correlated morphological change, may reflect a ‘line of least evolutionary resistance’ that manifests only in controlled environments, i.e., when there is no countervailing selection acting on cranial shape. To the best of our ability, we have also addressed the reviewers’ other concerns, including modifying and moving our figure highlighting the absence of phenotypic correlations between tibia length and residual cranial shape to the main text, in support of the finding that the cranial shape change is partially an indirect response to tibia selection.

We believe these revisions have greatly improved our manuscript, and we thank the editors and reviewers for their suggestions. We remain convinced that, with these revisions, our conclusions are robust and well supported by our data. We are only beginning to study candidate molecular mechanisms that drive increased growth of the Longshanks tibia. Without this information, we believe it would be risky to undertake functional analyses (e.g., knock‐out/in studies, growth plate in situs) with the goal of uncovering the genetic and molecular basis of the correlated change in the tibia and skull. We may undertake these studies in future, but they are outside the scope of this study, and are not necessary to support our interpretations.

[Editors’ note: further revisions were suggested prior to acceptance, as described below.]

Reviewer #1 :1. I think it would make sense for the authors to very briefly (a sentence or 2) address why they suggest why the basicranium and tibia are unrelated. Though I am not up on my mouse anatomy, I can easily see changes in for example tibia length, affecting muscle insertion points on the shoulder/neck area, which might also lead to correlated changes in muscles in this area then attached to the cranium – resulting in functional integration, of a sort. Reading the response to reviewer letter, this may have been taken out due to a reviewer concern, but I think something on this topic should be briefly included.

Thank you for this suggestion. While we cannot definitively rule out the possibility that changes in the cranium were related to functional integration with the tibia under selection, for example related to locomotion, we think this is not a likely explanation for the observed shape changes, for two reasons. First, although the Longshanks mouse does exhibit statistically significant changes in its gait in both the forelimb and hindlimb (Sparrow et al., 2017 *PeerJ* 5:e3008), the changes in limb joint angles (in the hind limb) were on the order of 3-5%, likely too small to exert any effect on muscle architecture and/or insertion points in other skeletal regions. Second, as a hind limb bone, the tibia has no direct musculotendinous connection with the pectoral girdle, cranial vertebrae, or cranium. With the exception of the biceps femoris, the most proximal muscles that insert onto the tibia originate on the pelvis (rectus femoris, hamstrings) or on the femur (vastus group), and insert on its proximal aspect. It is unlikely that changes to any of these muscles brought on by selection on tibia length would occasion changes in the axial skeleton that would be transferred from the pelvic region to the cranium across more than twenty articulations. We have added a brief clarification of this lack of anatomical connection between the two structures (lines 473 and 528).

2. Endochondral ossification – the authors should clarify why they suggest that a similar process through which the basicranium and long-bones develop would result in correlated evolution. This may have been in an earlier draft of the manuscript that was previously reviewed and removed, but I am a new reviewer and this is the first version I have seen. Citations supporting their reasoning would of course be welcome, I just think this should be made clear in the introduction – around Lines 62/65. Some of this may be a case of simply moving some sentences from then discussion to the introduction.

We have now addressed this suggestion on lines 89-97, as a rationale for the study. We argue that there is evidence from the developmental literature that numerous transgenic/KO mouse models show convergent phenotypic changes due to similar perturbations to the basicranium and limb bone growth plates. This suggests these structures are developmentally integrated via the action of pleiotropic genes, which we hypothesize could lead to their correlated evolution.